# Survival Outcome of Gastric Signet Ring Cell Carcinoma Based on the Optimal Number of Examined Lymph Nodes: A Nomogram- and Machine-Learning-Based Approach

**DOI:** 10.3390/jcm12031160

**Published:** 2023-02-01

**Authors:** Yongkang Lai, Junfeng Xie, Xiaojing Yin, Weiguo Lai, Jianhua Tang, Yiqi Du, Zhaoshen Li

**Affiliations:** 1Department of Gastroenterology, Ganzhou People’s Hospital Affiliated to Nanchang University, Ganzhou 341000, China; 2Department of Gastroenterology, Shanghai Changhai Hospital, Naval Medical University, Shanghai 200433, China

**Keywords:** gastric signet ring cell carcinoma, examined lymph nodes, minimal number, nomogram, machine learning

## Abstract

The optimal number of examined lymph nodes (ELNs) for gastric signet ring cell carcinoma recommended by National Comprehensive Cancer Network guidelines remains unclear. This study aimed to determine the optimal number of ELNs and investigate its prognostic significance. In this study, we included 1723 patients diagnosed with gastric signet ring cell carcinoma in the Surveillance, Epidemiology, and End Results database. X-tile software was used to calculate the cutoff value of ELNs, and the optimal number of ELNs was found to be 32 for adequate nodal staging. In addition, we performed propensity score matching (PSM) analysis to compare the 1-, 3-, and 5-year survival rates; 1-, 3-, and 5-year survival rates for total examined lymph nodes (ELNs < 32 vs. ELNs ≥ 32) were 71.7% vs. 80.1% (*p* = 0.008), 41.8% vs. 51.2% (*p* = 0.009), and 27% vs. 30.2% (*p* = 0.032), respectively. Furthermore, a predictive model based on 32 ELNs was developed and displayed as a nomogram. The model showed good predictive ability performance, and machine learning validated the importance of the optimal number of ELNs in predicting prognosis.

## 1. Introduction

Gastric cancer (GC) is the fifth most frequently diagnosed cancer and the third leading cause of cancer-related deaths worldwide [1]. Gastric signet ring cell carcinoma (GSRC) is an exceptional GC containing a large amount of mucus, accounting for 35–45% of new adenocarcinomas [2]. Compared with non-GSRC, GSRC has a good prognosis in the early stage, while it is characterized by low differentiation, high invasiveness, and poor prognosis in the advanced stage [3,4]. While significant progress has been made in understanding the pathogenesis and molecular biology of GSRC, adequate surgical resection and lymphadenectomy remain the mandatory backbone in treatment with curative intent. However, five-year overall survival after surgery is between 14 and 41.9% [5,6,7,8], and distant metastasis may be a primary reason. Therefore, adequate staging information will provide patients with a more accurate prognosis and precise treatment, improving patient survival to a certain extent.

Lymph node staging is crucial in determining the prognosis for GSRC. Several studies have demonstrated the relationship between the number of positive lymph nodes and long-term survival of GSRC patients [5,7,9,10]. However, in clinical practice, the number of positive lymph nodes is closely associated with the number of examined lymph nodes (ELNs), and a lower number of ELNs may result in fewer positive lymph nodes. Inaccurate lymph node staging may frequently occur due to inadequate intraoperative lymph node dissection, affecting the prognosis of patients and treatment decision-making. Therefore, determining the optimal number of ELNs is imperative to improve the prognosis for GSRC.

The optimal number of ELNs for GC has been extensively discussed in the past. According to the National Comprehensive Cancer Network (NCCN) guidelines, GC requires the examination of at least 16 ELNs [11]. According to Hu et al., a minimum of 32 ELNs should be examined in patients with pN3-stage GC [12]. Erstad et al. found that an ELN threshold ≥ 30 can significantly improve the prognosis of patients with GC [13]. However, as a more metastatic and more malignant type of GC, GSRC has rarely been studied, and the optimal number of ELNs for GSRC remains controversial. Therefore, the present study aimed to determine the optimal number of ELNs for GSRC; in addition, a prognostic model based on ELNs was developed for predicting the survival outcome for GSRC.

## 2. Patients and Methods

### 2.1. Data Source and Study Population

Due to the retrospective nature of the study, informed consent was not required. GSRC cases were obtained from the Surveillance, Epidemiology, and End Results (SEER) Program using SEER*Stat software (version 8.4.0, http://seer.cancer.gov/seerstat/, accessed on 16 May 2022) from the National Cancer Institute. The SEER database collects cancer diagnoses and survival data for about 30% of the US population and benefits from extensive quality review model development. Patients with GSRC diagnosed between 2010 and 2015 who underwent surgery plus lymph node dissection or sampling and had at least one lymph node harvested and examined according to the International Classification of Diseases in Oncology (ICD-O-3, histology code: 8490/3) were included in the study. The total number of regional lymph nodes examined was recorded by the pathologist. The other inclusion criteria were as follows: (1) patients with only one primary tumor; (2) ICD-O-3 identified the primary site as the stomach; and (3) patients with complete survival information.

The exclusion criteria were as follows: (1) unknown number of examined regional nodes; (2) unknown tumor grade, tumor size, or patient race; (3) unknown TNM or American Joint Committee on Cancer (AJCC) stage; (4) unknown surgical information; (5) age < 15 or >95 years old. The variables, including age, sex, race, grade, tumor size, examined lymph node count, tumor location, therapy, AJCC stage, and survival month, were collected from the SEER database. Finally, 1723 patients were included in the study; the detailed screening process for GSRC patients is depicted in Figure 1. Access to the SEER database did not require formal ethical approval, and its open-access policy is included.

### 2.2. Calculation of Cutoff Value for ELNs

The optimal number of ELNs was determined using X-tile plots. X-tile plots (version 3.6.1, Yale University School of Medicine, New Haven, CT, USA) can divide ELN data into low and high populations and assess all possible divisions [14]. The log-rank survival and means tests were utilized to calculate associations for each division. When a user double-clicked the mouse along the hypotenuse, the software automatically selected the point with the highest chi-square value as the best segmentation of the data. The enumeration method was used to calculate the number of lymph nodes from 15 to 35 for comparison and verify that the chi-square value at this point was the largest. The 1-, 3- and 5-year overall survival rates grouped by the optimal number of ELNs were compared after propensity score matching (PSM) analysis. The propensity model was subjected to PSM analysis in order to control and reduce selection bias and potential confounders. PSM analysis was calculated using a multivariate logistic regression model with the following covariates: age, sex, race, grade, size, site, therapy, and AJCC. The two groups were matched in a 2:1 ratio utilizing the nearest-neighbor method with a caliper width of 0.1. A value of *p* > 0.05 was considered a baseline balance between the two groups.

### 2.3. Model Development

Univariate and multivariate Cox regression analyses were used to select the predictive features. A value of *p* < 0.20 in the univariate analysis was incorporated into the multivariate analysis. The model was developed based on the optimal number of ELNs and features selected in multivariate analysis when *p* < 0.05. Finally, the optimal number of ELNs, age, treatment, AJCC, tumor size, and tumor site were used to construct a model and presented as a nomogram. To assess the predictive ability of this model in predicting the 1-, 3- and 5-year overall survival rates of patients, receiver operating characteristic curve (ROC) analysis was used, and the area under the ROC (AUC) was calculated. AUC values of 0.5 and 1.0 represented random chance and a significant ability to predict overall survival rates with the model. Calibration curves and decision-curve analysis (DCA) curves were used for the discrimination of the model and to determine the clinical net benefit associated with the use of the model. In addition, Extreme Gradient Boosting (XGBoost) was used to analyze the predictive ability with or without the optimal number of ELNs. XGBoost is a recently developed gradient tree boosting algorithm based on machine learning that combines the outputs of other decision trees to improve classification. XGBoost is scalable and allows faster calculations [15].

### 2.4. Statistical Analysis

The cutoff value for ELNs was calculated using X-tile software. The other statistical analyses were performed using R statistical software 4.2.0 (www.r-project.org, accessed on 6 July 2022). Continuous variables were expressed as mean ± standard deviation (SD) for normally distributed data, whereas for non-normally distributed data, continuous variables were expressed as medians (interquartile spacing). Categorical variables were expressed as proportions. The Kaplan–Meier method was used to calculate the survival data, while the log-rank test was further used to assess statistical significance. Variables with *p* < 0.05 were considered statistically significant.

## 3. Results

### 3.1. Baseline Characteristics

The data assembly process is depicted in Figure 1. A total of 4652 patients histologically diagnosed with GSRC between 2010 and 2015 were enrolled using the SEER database. Patients with an unknown number of regional nodes examined (n = 369), patients missing tumor grade, race, or tumor size (n = 1498), patients without TNM or AJCC stage information (n = 864), patients without surgical information (n = 148), and patients younger than 18 years old or older than 95 years old (n = 52) were excluded. Ultimately, 1723 patients were included in the study. These patients were randomly partitioned into a training cohort (n = 1206; 70%) and a validation cohort (n = 517; 30%). Detailed information is presented in Table 1. The median age was 63 years (IQR: 53–72), with 50.7% males. By race, white patients (66.8%) accounted for the majority of this study, while black and other patients accounted for 11.8% and 21.4%, respectively. The most common grade was Grade III (94.5%), and 653 (37.9) patients had a tumor size larger than 5 cm. Meanwhile, the gastric antrum was the most common site for GSRC (28.6%). Among all included patients, the 1-, 3-, and 5-year OS rates were 74.2%, 47%, and 29.6%, respectively.

### 3.2. Demarcation of the Minimum Number of ELNs

X-tile software determined 32 as the minimum number of ELNs. The Kaplan–Meier survival curves for OS showed that patients with ≥32 ELNs had better survival times than patients with <32 ELNs (*p* = 0.032, Figure 2). According to the NCCN guidelines, 16 or more ELNs are recommended. Hence, this study used the enumeration method to calculate the number of lymph nodes from 15 to 35 for comparison (Table 2). When 32 was calculated as the cutoff value for ELNs, the two groups had the maximum chi-square values and relative risk values and minimum *p*-values.

### 3.3. Baseline Comparisons on ELNs ≥ 32 and ELNs < 32 (Pre-PSM and Post-PSM)

To further verify 32 as the minimum number of ELNs, ELNs ≥ 32 and ELNs < 32 were analyzed as baselines for each group (Table 3). Before PSM, there were significant differences in five factors between the two groups, including race (*p* = 0.015), size (*p* = 0.002), site (*p* = 0.003), therapy (*p* = 0.001), and AJCC (*p* = 0.001). Moreover, patients with ≥32 ELNs had better 1-year survival rates than patients with <32 ELNs (80.2% vs. 72.9%, *p* = 0.01). However, the 3-year and 5-year survival rates between the two groups had no significant difference. After PSM, 548 patients with ≥32 ELNs were matched with 291 patients with <32 ELNs. There was no significant difference in the baseline characteristics between the two groups, indicating that each group’s selection bias was reduced and controlled. As shown in Table 3, patients with ≥32 ELNs had better 1-year survival rates (80.1% vs. 71.7%, *p* = 0.008), 3-year survival rates (51.2% vs. 41.8%, *p* = 0.09), and 5-year survival rates (30.2% vs. 27%, *p* = 0.032) than patients with <32 ELNs.

### 3.4. Development and Assessment of the Nomogram

To further verify that ELNs ≥ 32 can improve the prognosis of patients with GSRC, a nomogram was developed to predict the survival outcome in patients with GSRC based on the optimal number of ELNs and factors selected from Cox analysis. Table 4 shows the univariate and multivariate Cox analysis of factors influencing survival outcomes. The multivariate Cox analysis results revealed that age ≥ 75, size > 5 cm, ELNs < 32, tumor located in cardia, treatment with surgery only, and AJCC IV stage were associated with a poorer prognosis. The survival-prediction model of the nomogram was developed based on the above factors (Figure 3).

Figure 4a shows that the ROC curve of the nomogram predicted the 1-, 3-, and 5-year OS rate in the training cohort with AUCs of 0.790, 0.816, and 0.782, respectively. In the testing cohort (Figure 4b), the AUCs of the 1-, 3- and 5-year OS rates were 0.763, 0.809, and 0.804, respectively, indicating that the model had good predictive capability (Table 5). Figure 5 shows the calibration curve of the nomogram for predicting the OS rate at 1, 3, and 5 years, demonstrating good agreement between prediction and observation in the primary cohort (Figure 5a–c) and testing cohort (Figure 5d–f). DCA was used to compare nomogram clinical availability with AJCC staging. As illustrated in Figure 6, the *y*-axis represents the net benefit and the *X*-axis represents the probability of survival predicted by the model. DCA graphically showed that the nomogram was superior to AJCC staging under clinical conditions and that using a nomogram provided a greater net benefit to patients.

Table 5 and Figure 7 show predictive ability with or without the optimal number of ELNs using the XGBoost algorithm. The results showed that the XGBoost model including the optimal number of ELNs (n = 32) had better predictive ability than the XGBoost model excluding the optimal number of ELNs in predicting the OS rate at 1 (AUC: 0.803 vs. 0.753), 3 (AUC: 0.832 vs. 0.803), and 5 years (AUC: 0.806 vs. 0.792).

## 4. Discussion

Although the overall incidence of gastric cancer has decreased in recent decades, with the massive eradication of *Helicobacter pylori*, the incidence of GSRC is still increasing, especially in the young population [16,17]. As one of the most malignant types of GC, the prognosis for GSRC is closely associated with the stage. The prognosis for early GSRC is better than that for non-GSRC, while the prognosis for advanced GSRC is worse than for non-GSRC [18]. Furthermore, different stages have different treatment guidelines, which also induce differences in the survival prognosis of patients [2,11]. Therefore, determining accurate staging is crucial to improve the prognosis of patients with GSRC. The number of positive lymph nodes is a crucial factor for cancer staging. A high number of positive lymph nodes is also the most significant adverse prognostic indicator for GSRC because it provides a surrogate measure for both nodal burden and the likelihood of missed locoregional disease [13,19]. The number of ELNs provides information about the thoroughness of intraoperative lymph node clearance and is an accurate method for confirming the number of positive lymph nodes [20]. Adequate ELNs help determine prognosis and accurate staging, providing guidance for subsequent treatment and surveillance programs following treatment [11]. With inadequate ELNs, N1 and N2 patients may be misdiagnosed as N0 and receive different treatment and simpler monitoring procedures after treatment, resulting in a worse prognosis [1]. Furthermore, patients could have a higher risk of recurrence with inadequate ELNs, which might not cover the metastasized lymph nodes [20]. Therefore, in our study, we primarily focused on the number of ELNs and aimed to determine the optimal number of dissected lymph nodes.

According to the NCCN Clinical Practice Guidelines in Oncology (Version 2.2022), a minimum of 16 ELNs is recommended during GC surgery [11]. While it is true that patients with fewer than 16 examined lymph nodes may present a poorer prognosis, this value does not reflect the potential therapeutic value of expanded lymph node dissection for most patients [13]. Therefore, numerous studies have been carried out to identify the ELN threshold for survival advantage and oncologic benefit. For example, Smith et al. indicated that every increase of 10 lymph node dissections increases patient survival by approximately 7%, and this survival advantage can be sustained up to a threshold of 40 lymph node dissections [21]. Brenkman et al. recommended a minimum of 25 ELNs for prolonged survival [22], while Ichikura et al. observed that a minimum of 30 ELNs was associated with improved survival for patients with advanced-stage GC [23]. However, their recommended thresholds varied significantly, which was likely to affect clinical practice, and it is not appropriate to apply the same standards to GSRC, which is more malignant and prone to metastasis.

Currently, there are limited studies identifying the optimal number of ELNs for GSRC. Therefore, it is clinically important to investigate the optimal number of ELNs for GSRC. In our study, we analyzed patients from 2010 to 2015 in the SEER database and confirmed that 32 was the adequate cutoff number for lymph node dissection. We also confirmed that prognosis would be improved with ELNs > 32. Data from 2010 to 2015 is more comprehensive, including the availability of the complete 7th-edition AJCC classification, in addition to more complete 5-year survival data. The X-title software was used to confirm 32 as the optimal cutoff value. The chi-square values and relative risk were listed when the number of lymph nodes examined was set at 15–35. When 16 or 25 was used as the cutoff point in our modeling data, the results showed no difference in prognosis. However, when 30 was used as the cutoff value, although the *p*-value showed significance, the chi-square value and the relative risk were both less than when 32 was used as the critical value. Moreover, we compared 1-, 3-, and 5-year survival rates in groups of 32, and developed a clinical predictive model based on 32 ELNs; all of these showed that a minimum of 32 ELNs for GSRC patients could improve the survival rate. Additionally, the results of the XGBoost algorithm showed that the model including the optimal number of ELNs had better predictive ability than the model excluding the optimal number of ELNs in predicting the OS rate for patients with GSRC. Although there was little difference in the predictive ability for 5-year survival, a significant difference was seen in the predictive ability of the XGBoost model containing the optimal number of ELNs at 1 and 3 years. This may be an issue with the data sample, and future expansion of the data sample may reveal significant differences in 5-year survival rates. Therefore, we considered the minimum of 32 ELNs to have a wider practical application for GSRC patients. Some believe that additional resection of ELNs can increase the difficulty of surgery and the potential risk of complications, resulting in a negative prognosis. However, previous studies including large samples have shown that ELNs in patients with gastric cancer ≥ 30 can significantly improve survival prognosis [13,21,23,24,25]. Furthermore, the 8th edition of the AJCC Cancer Staging Manual highlights that the removal of ≥30 nodes is desirable for patients with gastric cancer, which is similar to our results [26]. Therefore, we believe that additional ELNs do not worsen the prognosis of patients and would contribute to more completed tumor clearance and better staging.

Interestingly, our results are also similar to previous studies exploring the optimal number of ELNs for GC [12,13,23,27]. Among these, a large cohort study conducted by Huang et al. by analyzing two databases of patients with non-metastatic gastric adenocarcinoma found the minimum number of ELNs and the optimal number of ELNs for gastric adenocarcinoma to be 17 and 33, respectively [27]. Accordingly, we could venture a guess that the optimal number of ELNs for GC might be concentrated around that level. Therefore, we suggest that the number of ELNs should be at least greater than 32 in patients with surgically resectable GC. However, the current studies in this area are still primarily retrospective data analyses, and there is definitely a need to design higher-quality multicenter randomized controlled studies to investigate the optimal number of ELNs in GC more accurately in the future.

There are certain limitations in the current study. First, this is a retrospective study, which may include biases and potential confounding variables. Second, this study only focused on the number of ELNs and ignored the location of lymph nodes, which may be another important factor affecting the prognosis, for which a further study will be conducted in the future. Third, given the characteristics of the data relied on in our own study, the results of this study may be more applicable to patients who did not receive neoadjuvant chemotherapy due to the inability to differentiate between chemotherapy information, but this study still has reference value for other patients with GSRC treated with chemotherapy (either neoadjuvant or adjuvant). In future studies based on our own unit center, we will further differentiate chemotherapy and explore the effects of neoadjuvant, adjuvant, and neoadjuvant + adjuvant subgroups on the stability of the model to fully validate and improve the model and make the prediction results more stable. Fourth, the data for this study were obtained from authoritative public databases whose data standards were all uniform, but the data did in turn come from various subcenters. Therefore, there may be diversity in the way lymph nodes were examined and processed. However, the data entered into the database are worthy of recognition, and the results of our study still have reference value. Fifth, the selected SEER cases do not cover all states in the United States, and the applicability to Asian populations needs further study. Therefore, future validation of this study in Asian patients with GSRC is needed. However, this study included a large sample size of GSRC and was the first to investigate the optimal number of ELNs to be cleared for GSRC. Hence, this study is still of significant clinical relevance.

In conclusion, 32 was demonstrated to be the optimal number of ELNs for adequate nodal staging, and the prognostic model based on the optimal number of ELNs also showed good performance in predictive ability.

## Figures and Tables

**Figure 1 jcm-12-01160-f001:**
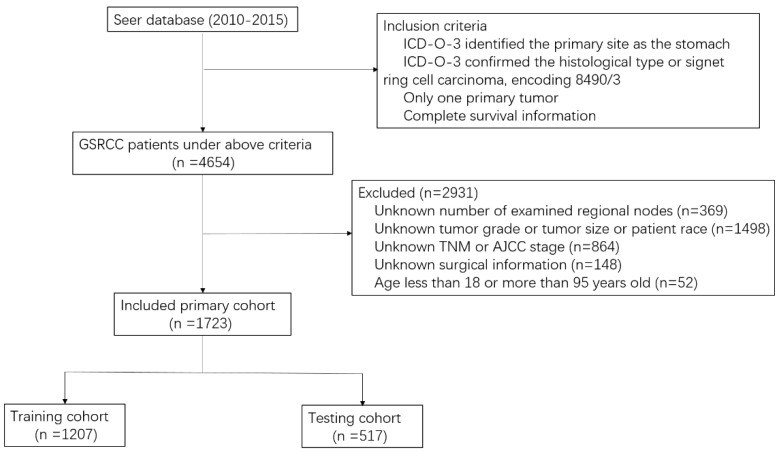
Flowchart of patients included in the present study.

**Figure 2 jcm-12-01160-f002:**
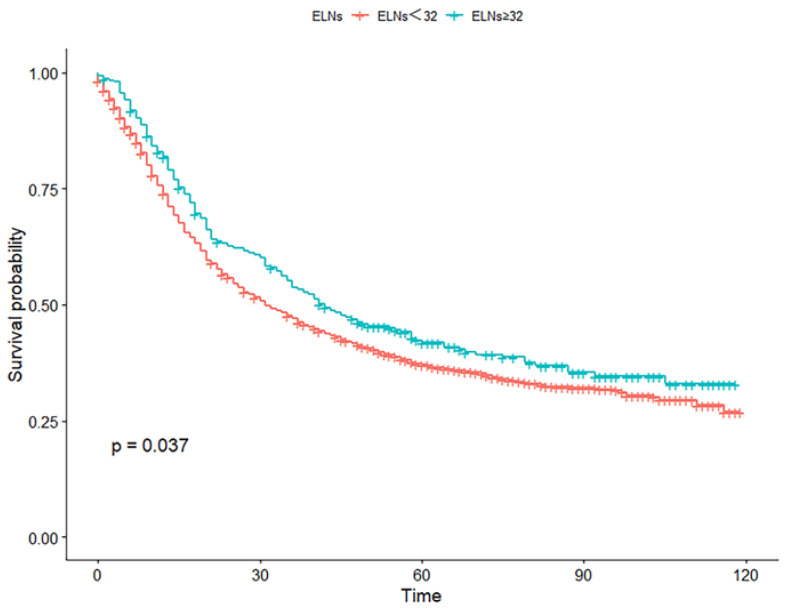
Kaplan–Meier survival curves for overall survival in patients with ELNs < 32 and ELNs ≥ 32. ELNs, examined lymph nodes.

**Figure 3 jcm-12-01160-f003:**
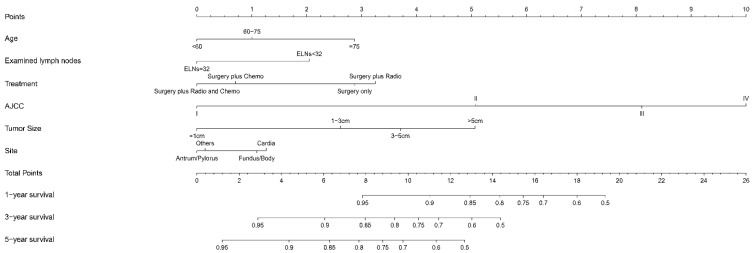
Nomogram based on examined lymph nodes for predicting survival outcome in patients with gastric signet ring cell carcinoma.

**Figure 4 jcm-12-01160-f004:**
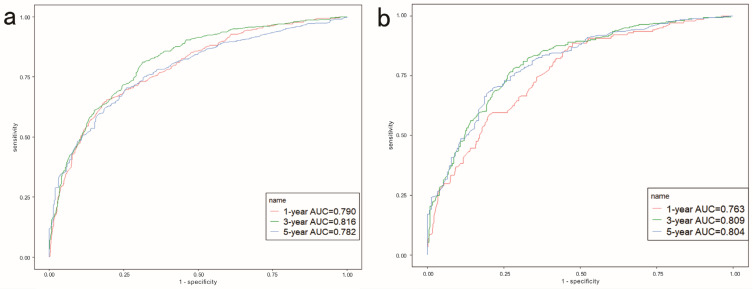
The area under the receiver operating characteristic curve (AUROC) of the model in predicting 1-, 3-, and 5-year overall survival (OS) in the training cohort (**a**) and 1-, 3-, and 5-year OS in the validation cohort (**b**).

**Figure 5 jcm-12-01160-f005:**
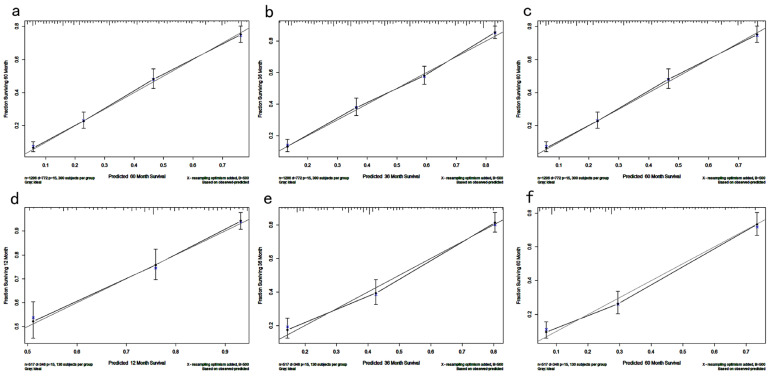
Calibration curves of the model for predicting 1- (**a**), 3- (**b**), and 5-year OS (**c**) in the training cohort and 1- (**d**), 3- (**e**), and 5-year (**f**) OS in the validation cohort.

**Figure 6 jcm-12-01160-f006:**
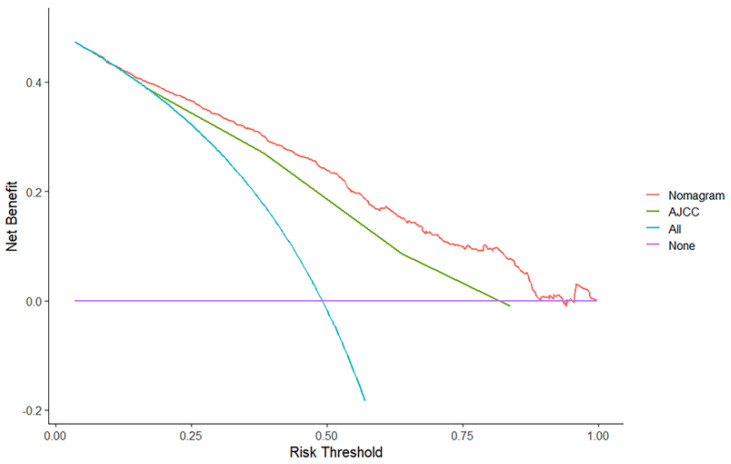
Decision curve analysis for the nomogram.

**Figure 7 jcm-12-01160-f007:**
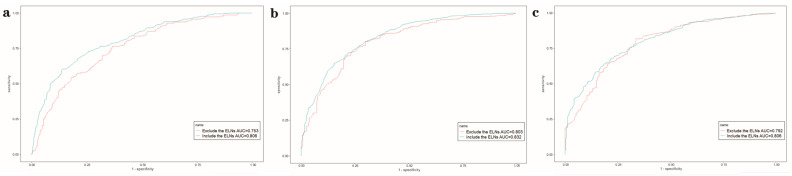
The area under the receiver operating characteristic curve (AUROC) of the XGBoost model for predicting 1- (**a**), 3- (**b**), and 5- (**c**) year overall survival with or without the optimal number (n = 32) of examined lymph nodes (ELNs).

**Table 1 jcm-12-01160-t001:** Demographic and clinicopathological characteristics of patients.

Variables	All Patients (n = 1723)	Training Cohort (n = 1206)	Validation Cohort (n = 517)
Age (median, IQR)	63 (53–72)	63 (52–72)	62 (52–72)
Sex [No. (%)]			
Male	873 (50.7)	613 (50.8)	260 (50.3)
Female	850 (49.3)	593 (49.2)	257 (49.7)
Race [No. (%)]			
White	1151 (66.8)	797 (66.1)	354 (68.5)
Black	203 (11.8)	145 (12)	58 (11.2)
Other	369 (21.4)	264 (21.9)	105 (20.3)
Grade [No. (%)]			
I	2 (0.1)	1 (0.1)	1 (0.2)
II	45 (2.6)	32 (2.7)	13 (2.5)
III	1628 (94.5)	1141 (94.5)	487 (94.2)
IV	48 (2.8)	32 (2.7)	16 (3.1)
Size [No. (%)]			
≤1 cm	117 (6.8)	85 (7)	39 (7.5)
1–2 cm	249 (14.5)	173 (14.3)	80 (15.5)
2–3 cm	267 (15.5)	186 (15.4)	74 (14.3)
3–4 cm	236 (13.7)	168 (13.9)	76 (14.7)
4–5 cm	201 (11.6)	148 (12.3)	65 (12.6)
>5 cm	653 (37.9)	446 (37.1)	183 (35.4)
Examined lymph node count (year, mean ± SD)	20.4 ± 0.33	20.3 ± 0.39	20.7 ± 0.62
Site [No. (%)]			
Cardia	239 (13.9)	165 (13.7)	74 (14.3)
Fundus of stomach	42 (2.4)	34 (2.8)	8 (1.6)
Body of stomach	207 (12)	143 (11.9)	64 (12.4)
Gastric antrum	492 (28.6)	342 (28.4)	150 (29)
Pylorus	91 (5.3)	71 (5.9)	20 (3.9)
Lesser curvature of stomach	228 (13.2)	158 (13.1)	70 (13.5)
Greater curvature of stomach	102 (5.9)	73 (6.1)	27 (5.2)
Overlapping	177 (10.3)	115 (9.5)	62 (12)
Stomach NOS	145 (8.4)	103 (8.6)	42 (8.1)
Therapy [No. (%)]			
Surgery only	633 (36.7)	447 (37.1)	186 (36)
Surgery plus radio	22 (1.3)	14 (1.2)	8 (1.5)
Surgery plus chemo	517 (30)	354 (29.3)	163 (31.5)
Surgery plus cadio and chemo	551 (32)	391 (32.4)	160 (31)
AJCC [No. (%)]			
I	379 (22)	273 (22.6)	106 (20.5)
II	374 (21.7)	268 (22.2)	106 (20.5)
III	785 (45.6)	540 (44.8)	245 (47.4)
IV	185 (10.7)	125 (10.4)	60 (11.6)
Overall survival [No. (%)]			
1-year	1278 (74.2)	902 (74.8)	376 (72.7)
3-year	810 (47)	577 (47.8)	233 (45.1)
5-year	510 (29.6)	355 (29.4)	155 (30)

**Table 2 jcm-12-01160-t002:** Analysis of the cutoff value for ELNs.

Cut off Value for ELNs	Chi-Square Score	Relative Risk	*p* Value
<15 vs. ≥15	3.02	1.05	0.18
<16 vs. ≥16	2.92	1.04	0.083
<17 vs. ≥17	3.67	1.05	0.088
<18 vs. ≥18	3.53	1.06	0.056
<19 vs. ≥19	5.35	1.07	0.061
<20 vs. ≥20	4.21	1.06	0.021
<21 vs. ≥21	4.45	1.06	0.041
<22 vs. ≥22	3.96	1.05	0.035
<23 vs. ≥23	3.53	1.05	0.047
<24 vs. ≥24	2.51	1.03	0.58
<25 vs. ≥25	4.39	1.05	0.11
<26 vs. ≥26	3.79	1.05	0.037
<27 vs. ≥27	3.06	1.05	0.052
<28 vs. ≥28	3.4	1.06	0.081
<29 vs. ≥29	4.04	1.07	0.066
<30 vs. ≥30	3.44	1.07	0.045
<31 vs. ≥31	4.35	1.09	0.063
<32 vs. ≥32	5.45	1.11	0.037
<33 vs. ≥33	3.91	1.10	0.02
<34 vs. ≥34	4.69	1.10	0.048
<35 vs. ≥35	3.48	1.10	0.03

**Table 3 jcm-12-01160-t003:** Clinical characteristics of patients in the study before and after propensity score matching.

Characteristics	Before Matching			After Matching		
	ELN < 32 (n = 1430)	ELN ≥ 32 (n = 293)	*p*	ELN < 32 (n = 548)	ELN ≥ 32 (n = 291)	*p*
Age (median, IQR)	63 (52–73)	60 (50–68)		60 (51–71)	60 (50–68)	0.121
Sex [No. (%)]			0.843			0.640
Male	723 (50.6)	150 (51.2)		288 (52.6)	148 (52.9)	
Female	707 (49.4)	143 (48.8)		260 (47.4)	143 (49.1)	
Race [No. (%)]			0.015			0.519
White	955 (66.8)	196 (66.9)		359 (65.5)	194 (66.7)	
Black	181 (12.7)	22 (7.5)		72 (13.1)	22 (7.5)	
Other	294 (20.5)	75 (25.6)		117 (21.4)	75 (25.8)	
Grade [No. (%)]			0.789			0.988
I	2 (0.1)	0		0	0	
II	39 (2.7)	6 (2)		9 (1.6)	5 (1.7)	
III	1348 (94.3)	280 (95.6)		525 (95.8)	279 (95.9)	
IV	41 (2.9)	7 (2.4)		14 (2.6)	7 (2.4)	
Size [No. (%)]			0.002			0.485
≤1 cm	107 (7.5)	10 (3.4)		24 (4.4)	10 (3.4)	
1–2 cm	212 (14.8)	37 (12.6)		53 (9.7)	35 (12)	
2–3 cm	225 (15.7)	42 (14.3)		65 (11.8)	42 (14.4)	
3–4 cm	205 (14.3)	31 (10.6)		77 (14.1)	31 (10.7)	
4–5 cm	169 (11.8)	32 (10.9)		68 (12.4)	32 (11)	
>5 cm	512 (35.9)	141 (48.2)		261 (47.6)	141 (48.5)	
Site [No. (%)]			0.003			0.064
Cardia	210 (14.7)	29 (9.9)		69 (12.6)	28 (9.6)	
Fundus of stomach	34 (2.4)	8 (2.7)		10 (1.8)	8 (2.7)	
Body of stomach	159 (11.1)	48 (16.4)		60 (10.9)	47 (16.2)	
Gastric antrum	427 (29.9)	65 (22.2)		155 (28.3)	65 (22.3)	
Pylorus	81 (5.7)	10 (3.4)		31 (5.7)	10 (3.4)	
Lesser curvature of stomach	182 (12.7)	46 (15.7)		74 (13.6)	46 (15.8)	
Greater curvature of stomach	84 (5.9)	18 (6.1)		27 (4.9)	18 (6.2)	
Overlapping	141 (9.8)	36 (12.3)		78 (14.2)	36 (12.4)	
Stomach NOS	112 (7.8)	33 (11.3)		44 (8)	33 (11.4)	
Therapy [No. (%)]			0.001			0.130
Surgery only	555 (38.8)	78 (26.6)		161 (29.4)	76 (26.1)	
Surgery plus radio	18 (1.3)	4 (1.4)		7 (1.3)	4 (1.4)	
Surgery plus chemo	391 (27.3)	126 (43)		175 (31.9)	126 (43.3)	
Surgery plus radio and chemo	466 (32.6)	85 (29)		205 (37.4)	85 (29.2)	
AJCC [No. (%)]			0.001			0.469
I	329 (23)	50 (17)		81 (14.8)	48 (16.5)	
II	326 (22.8)	48 (16.4)		113 (20.6)	48 (16.5)	
III	621 (43.4)	164 (56)		304 (55.5)	164 (56.4)	
IV	154 (10.8)	31 (10.6)		50 (9.1)	31 (10.6)	
Overall survival [No. (%)]						
1-year	1043 (72.9)	235 (80.2)	0.01	393 (71.7)	233 (80.1)	0.008
3-year	659 (46.1)	151 (51.5)	0.088	229 (41.8)	149 (51.2)	0.009
5-year	420 (29.4)	90 (30.7)	0.646	148 (27)	88 (30.2)	0.032

**Table 4 jcm-12-01160-t004:** Univariate and multivariable analyses in the derivation cohort.

Variable	n	Univariate Analyses	Multivariable Analyses		
		*p*	HR	CI	*p*
Age					
<60	523 (43.4)	Reference	-	-	Reference
60–75	481 (39.9)	0.002	1.251	1.065–1.471	0.006
≥75	202 (16.7)	<0.001	1.898	1.543–2.335	<0.001
Sex					
Male	613 (50.8)	Reference			
Female	593 (49.2)	0.268			
Race					
White	797 (66.1)	Reference			
Black	145 (12)	0.471			
Other	264 (21.9)	0.216			
Grade					
I	1 (0.1)	Reference			
II	32 (2.7)	0.819			
III	1141 (94.6)	0.82			
IV	32 (2.7)	0.823			
Size					
≤1 cm	85 (7)	Reference	-	-	Reference
1–3 cm	359 (29.8)	<0.001	1.813	1.097–2.996	0.02
3–5 cm	316 (26.2)	<0.001	2.312	1.382–3.868	0.001
>5 cm	446 (37)	<0.001	3.131	1.867–5.251	<0.001
ELNs					
<32	1004 (83.3)	Reference	-	-	Reference
≥32	202 (16.7)	0.014	0.633	0.518–0.775	<0.001
Site					
Others	451 (37.4)	Reference	-	-	Reference
Cardia	165 (13.7)	0.362	1.282	1.017–1.617	0.036
Fundus/Body	177 (14.7)	0.633	1.234	0.981–1.533	0.072
Antrum/Pylorus	413 (34.2)	0.042	0.967	0.729–1.282	0.814
Therapy					
Surgery only	447 (37.1)	Reference	-	-	Reference
Surgery plus radio	14 (1.2)	0.006	1.090	0.617–1.926	0.767
Surgery plus chemo	354 (29.4)	0.012	0.619	0.508–0.754	<0.001
Surgery plus radio and chemo	391 (32.4)	0.585	0.529	0.434–0.644	<0.001
AJCC					
I	273 (22.6)	Reference	-	-	Reference
II	268 (22.2)	<0.001	3.127	2.308–4.235	<0.001
III	540 (44.8)	<0.001	6.150	4.577–8.264	<0.001
IV	125 (10.4)	<0.001	9.373	6.701–13.111	<0.001

**Table 5 jcm-12-01160-t005:** Prognostic efficiency of the nomogram of OS.

		AUC		
	C-Index (Internal Validation)	1-Year Survival	3-Year Survival	5-Year Survival
Training cohort	0.748	0.790	0.816	0.782
Testing cohort	-	0.763	0.809	0.804
XGBoost				
Including ELNs		0.803	0.832	0.806
Excluding ELNs		0.753	0.803	0.792

## Data Availability

Data available on request due to restrictions e.g., privacy or ethical.

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
