# Peer review of "Survival Outcome of Gastric Signet Ring Cell Carcinoma Based on the Optimal Number of Examined Lymph Nodes: A Nomogram- and Machine-Learning-Based Approach"

_jcm, 2023, doi:10.3390/jcm12031160_

Round 1
Reviewer 1 Report
Dear Authors,
Thank you for submitting your paper. This is a well thought paper with interesting results. I believe it could be improved in regards to its discussion section, which seems to be too basic taking into account your findings.
In my view, there is a fundamental issue in your data. You present data for different types of therapy (surgery alone, surgery plus radio, surgery plus chemo and surgery plus radio and chemo) but you do not state when was chemo and radio done, if this was done neoadjuvantly ou adjuvantly. This is, in my view, extremely important as you claim, in your discussion, that you believe the difference in survival between groups ENL<32 and ENL>32 can be due to " ...inadequate ELNs, N1 and N2 patients could be misdiagnosed as N0, who would receive different treatment and simpler monitoring procedures after treatment, resulting in a worse prognosis". Accordingly to NCCN guidelines, patients receiving neoadjuvant chemotherapy should receive chemotherapy adjuvantly as well, and this is not dependent on histologic N status. Your remark would only be true for patients not receiving chemotherapy neoadjuvantly and even for those, they would have to be >pT2 to not receive chemotherapy anyway. For this reason it is important to know how many patients actually received neoadjuvant therapy so you can state that the benefit of survival is due to better identification of N1 patients. There are some papers stating the benefit of survival of a more extended lymphadenectomy even in N0 patients, which is a very interesting field of study and not yet clearly understood. Your results are very interesting but their interpretation can be different, and this should be discussed.
We can also see in your table 3 that, even though there was no statistically significant difference between the different types of therapy given between the groups ELN<32 and ELN>32, there is a remarkable difference in the number of patients that received chemotherapy 31.9% (ELN<32) vs 43.3% (ELN>32). Taking into account that the use of chemotherapy (even more than the use of radiochemotherapy) has a significant influence in distant micrometastasis, and that this fact may be of importance for differences in survival, I believe this should be addressed and discussed.
Other issue with this theme is that the number of lymph nodes in the specimen can be found after lots of different processing protocols. Some groups perform the lymph nodes dissection of the specimen ex-vivo by the surgeons in theatre, some depend on pathologists dissection ability and some process every part of the fat tissue chasing for nodes. Obviously the number of lymph nodes will be very different depending on the protocol they use. This database most likely has data taken from groups using all kinds of protocols to process the lymph nodes, which is a confounding factor and probably should be stated in your discussion.
You used a confusing way of present the references numbers in your manuscript, as it does not start with number 1 and does not go incrementing numbers as soon as the references appear in the text. For instance your first reference in the text is number 16. This is confusing and should be addressed.
Line 12 - "The National Comprehensive Cancer Network guidelines for gastric signet ring cell carcinoma recommend that surgeons’ samples remain unclear." - What do you mean with this?
Line 14 - A total of 1723 patients diagnosed with gastric signet ring cell carcinoma
Line 16 - The cut-off value of lymph nodes was calculated using X-tile software
Line 19 - The optimal number of ELNs for adequate nodal staging was found to be 32
Line 46 - "Inaccurate lymph node staging may frequently occur due to inadequate intraoperative lymph node examination" - Do you mean lymph node clearance/dissection or do you mean an examination by the pathologist in theatre?
Line 55 - ...and apparently, the optimal number of ELNs of GSRC remain controversial
Line 61 - unknown number of examined regional nodes; (2) unknown tumor grade or tumor size or patient race; (3) unknown TNM or the American Joint Committee on Cancer (AJCC) stage; (4) unknown surgical information;
Line 128 - Moreover, patients with unknown number of regional nodes examined
Line 140 - In your flow chart (figure 1) the excluded patients' box should have proper grammar - should be: unknown number of examined regional nodes, unknown tumor grade or tumor size or patient race, unknown TNM or the American Joint Committee on Cancer (AJCC) stage.
Line 142 - Table 1, several variables present percentages that do not sum 100%. Some sum 100.1% others 99%. This should be corrected in order to all variables percentages sum 100%.
Line 168 - Table 3, several variables present percentages that do not sum 100%. Some sum 100.1% others 99%. This should be corrected in order to all variables percentages sum 100%.
Line 175 - "The multivariate Cox analysis results revealed that age ≥ 75, size >5 cm, ELNs <32, tumor located in antrum/pylorus... were associated with a poorer prognosis" - The fact that you found that patients having more distal tumors (antrum and pylorus) were associated with poorer prognosis is surprising and should be discussed, why do you believe that happened?
Line 175 - "... treatment with surgery only...
Line 204 - The results showed that the XGBoost model that included the optimal number of ELNs (n = 32) had better predictive ability than the XGBoost model that excluded the optimal number of ELNs...
Line 208 - These pictures (figure 7) cannot be interpreted as the pictures are presented with a very poor definition - should be improved.
Line 225 - clearance and is an accurate method
Line 226 - Adequate ELNs help determine prognosis
Line 233 - what do you mean with "intra-operative ELNs"? Do you want to say the optimal number of dissected lymph nodes? Or the number of lymph nodes examined by the pathologist in theatre?
Line 236 - ...with fewer than 16 examined lymph nodes may present a poorer prognosis...
Line 247 - ... Currently, there are limited studies identifying the optimal number...
Line 250 - "...confirmed that 32 was the adequate lymph node a cut-off number for dissection, while prognosis would not be improved with ELNs >32." - This sentence is confusing, firstly should be reviewed and rewritten with a better grammar, secondly, from what I gathered, you showed exactly the opposite, that prognosis (survival) was improved with ELNs >32.
Line 261 - " Finally, we used the XGBoost model compared the predict ability included the optimal number of ELNs of GSRC and excluded the optimal number of ELNs." This is a very confusing sentence and should be reviewed.
Line 265 - What do you mean with "the best ELNs"?
Line 268 - However, some may believe...
Line 274 - you mention a "large randomized clinical trial in the United States has demonstrated that systematic lymph node sampling would only prolong operative time and increase bleeding rate but has no effect on survival and does not increase postoperative hospital stay or the incidence of postoperative complications". However, your references related to this statement are 2 papers reviewing lymph node dissection for lung cancer. I believe you should not infer about abdominal lymphadenectomy presenting references of papers discussing mediastinal and hilar lymph node dissection in lung cancer.
Line 279 - For which a further study will be conducted in the future.
Line 282 - Asian population
Line 282 - "gastric IMC" - Please present the meaning of abbreviations the first time you use them in the text.
Author Response
- In my view, there is a fundamental issue in your data. You present data for different types of therapy (surgery alone, surgery plus radio, surgery plus chemo and surgery plus radio and chemo) but you do not state when was chemo and radio done, if this was done neoadjuvantly ou adjuvantly. This is, in my view, extremely important as you claim, in your discussion, that you believe the difference in survival between groups ENL<32 and ENL>32 can be due to " ...inadequate ELNs, N1 and N2 patients could be misdiagnosed as N0, who would receive different treatment and simpler monitoring procedures after treatment, resulting in a worse prognosis". Accordingly to NCCN guidelines, patients receiving neoadjuvant chemotherapy should receive chemotherapy adjuvantly as well, and this is not dependent on histologic N status. Your remark would only be true for patients not receiving chemotherapy neoadjuvantly and even for those, they would have to be >pT2 to not receive chemotherapy anyway. For this reason it is important to know how many patients actually received neoadjuvant therapy so you can state that the benefit of survival is due to better identification of N1 patients. There are some papers stating the benefit of survival of a more extended lymphadenectomy even in N0 patients, which is a very interesting field of study and not yet clearly understood. Your results are very interesting but their interpretation can be different, and this should be discussed.
Response: In this study, the data we used were obtained from the authoritative public database the Surveillance, Epidemiology, and End Results (SEER) database, using SEER*Stat software (version 8.4.0, http://seer.cancer.gov/seerstat/) from the National Cancer In-stitute. Unfortunately, in this database, only chemotherapy history is provided and information on the timing of chemotherapy and radiotherapy is missing, so this study cannot provide information on whether the treatment was neoadjuvant or adjuvant or both. Therefore, your concern about neoadjuvant and adjuvant chemotherapy and its timing cannot be shown. In this regard, we searched literature published in SCI using SEER database for similar studies, such as prognosis of bladder cancer (Front Oncol 12 (2022):789028) based on the number of lymph nodes, prognosis of lung cancer (Transl Lung Cancer Res 10(2021):815-825), prognosis of esophageal cancer (Ann Surg Oncol 27(2020):2042-2050), etc. They also did not differentiate between chemotherapy information, but did not affect the reference value of such studies. Besides, gastric signet ring cell carcinoma (GSRC) is a particular histological subtype of gastric adenocarcinomas displaying a worse prognosis. Even though the perioperative chemotherapy in resectable gastric adenocarcinomas demonstrated a significant benefit in terms of overall survival compared to surgery alone (N Engl J Med. 355 (2006):11–20; J Clin Oncol 29 (2011):1715–21), this benefit seems to be limited to non-GSRC histology (Ann Surg 254 (2011):684–93). And previous studies have suggested GSRC may have inherent chemo resistance (Eur J Cancer 30A (1994):1263–1269; Ann Surg 250 (2009):878–887; Oncol Rep 7 (2000):841–846; Clin Cancer Res 14 (2008):2012–2018), the role of chemotherapy in GSRC is controversial, especially the limited effect on positive lymph nodes. Furthermore, currently, no specific recommendation is available about the type of lymphadenectomy to perform for GSRC, and this study is the first study to investigate the optimal number of examed lymph nodes intraoperatively in GSRC. Although information on the timing of radiotherapy and chemotherapy is lacking, it is still useful for clinical practice and future studies. It needs to be acknowledged that if the study includes the information about neoadjuvant and adjuvant will further improve the application of the model constructed in this study. In this regard, we truthfully point out this flaw in the Discussion section of this study to avoid misinformation. In summary, we believe that the innovative elements of our work are more clearly emphasized, and hopefully that you could satisfactorily respond to this comment, which helped to improve the quality of the manuscript.
- We can also see in your table 3 that, even though there was no statistically significant difference between the different types of therapy given between the groups ELN<32 and ELN>32, there is a remarkable difference in the number of patients that received chemotherapy 31.9% (ELN<32) vs 43.3% (ELN>32). Taking into account that the use of chemotherapy (even more than the use of radiochemotherapy) has a significant influence in distant micrometastasis, and that this fact may be of importance for differences in survival, I believe this should be addressed and discussed.
Response: Thank you for this helpful suggestion. As shown in the study, the number of cases with and without chemotherapy did differ under the threshold cut-off value of 32 lymph nodes, with a difference of 43%-32%=11%, which definitely indicates that chemotherapy has a certain effect, but 11% does not indicate the size of the effect of chemotherapy, which needs to be analyzed by regression analysis and regression coefficients, and the chi-square test can only indicate whether there is a difference between the two. However, this threshold for the number of lymph nodes is the optimal cut-off threshold that we calculated using the well-known software X-title based on the actual prognosis of the patients. Therefore, in such a case, the differences between the 2 groups are not necessarily all due to chemotherapy factors, and may also be related to the fact that this study is a retrospective study design. Actually, the timing of chemotherapy has influence in the survival of patients with GSRC, therefore, for which a further study will be conducted in the future, and we will discuss it in the Discussion section according to your advice.
- Other issue with this theme is that the number of lymph nodes in the specimen can be found after lots of different processing protocols. Some groups perform the lymph nodes dissection of the specimen ex-vivo by the surgeons in theatre, some depend on pathologists dissection ability and some process every part of the fat tissue chasing for nodes. Obviously the number of lymph nodes will be very different depending on the protocol they use. This database most likely has data taken from groups using all kinds of protocols to process the lymph nodes, which is a confounding factor and probably should be stated in your discussion.
Response: Thank you for this very helpful comment. The data for this study were obtained from authoritative public databases whose data standards were all uniform, but the data did in turn come from various subcenters. Therefore, there may be diversity in the way lymph nodes are taken. However, the data entered into the database are worthy of recognition. We strongly agree with you that this may be a shortcoming of studies relying on public databases. In this regard, we have discussed it the manuscript according to your advice.
- You used a confusing way of present the references numbers in your manuscript, as it does not start with number 1 and does not go incrementing numbers as soon as the references appear in the text. For instance your first reference in the text is number 16. This is confusing and should be addressed.
Response: Thank you for this helpful suggestion. We are sorry that we have confused the format of the references, and we have corrected the references numbers according to your advice.
- Line 12 - "The National Comprehensive Cancer Network guidelines for gastric signet ring cell carcinoma recommend that surgeons’ samples remain unclear." - What do you mean with this?
Response: Thank you for this careful comment. We are sorry that it maybe our mistake that we did not express this sentence clearly. The National Comprehensive Cancer Network guidelines has recommended the optimal number of examined lymph nodes the for gastric cancer, however, the optimal number of examined lymph nodes the for gastric signet ring cell carcinoma remains unclearly. We have revised it in the manuscript according to your advice, as shown in Abstract section. Finally, we sincerely appreciate your valuable suggestions on our manuscript. According to these valuable suggestions, we have substantially edited our manuscript, and the quality of the manuscript has been greatly improved.
- Line 14 - A total of 1723 patients diagnosed with gastric signet ring cell carcinoma
Response: Thank you for this helpful comment. We have corrected this sentence to “In this study, we included 1723 patients diagnosed as gastric signet ring cell carcinoma in the Surveillance, Epidemiology, and End Results database” according to your advice in the Abstract section.
- Line 16 - The cut-off value of lymph nodes was calculated using X-tile software
Response: Thank you for this advice. We have corrected this sentence to “X-tile software was usedcalculated the cut-off value of ELNs” according to your advice in the Abstract section.
- Line 19 - The optimal number of ELNs for adequate nodal staging was found to be 32
Response: Thank you for this comment. We have revised it according to your advice in the Abstract section.
- Line 46 - "Inaccurate lymph node staging may frequently occur due to inadequate intraoperative lymph node examination" - Do you mean lymph node clearance/dissection or do you mean an examination by the pathologist in theatre?
Response: Thank you for this careful comment. We are sorry that we did not express it clearly. In this sentence, we mainly want to explain the importance of lymph node clearance/dissection in surgery. As we all know, the number of examed lymph nodes informed the thoroughness of clearance, which helped judging prognosis and accurate staging while providing guidance for adjuvant treatment and surveillance programs following treatment. Therefore, inadequate lymph node clearance/dissection will result in inaccurate lymph node staging, which may have a big impact on the prognosis of patients. We have revised the sentence in the manuscript according to your advice.
- Line 55 - ...and apparently, the optimal number of ELNs of GSRC remain controversial
Response: Thank you for this comment. We have revised it in the manuscript according to your advice.
- Line 61 - unknown number of examined regional nodes; (2) unknown tumor grade or tumor size or patient race; (3) unknown TNM or the American Joint Committee on Cancer (AJCC) stage; (4) unknown surgical information;
Response: Thank you for this comment. We have revised it in the manuscript according to your advice.
- Line 128 - Moreover, patients with unknown number of regional nodes examined
Response: Thank you for your advice. We have revised this sentence in the manuscript according to your advice.
- Line 140 - In your flow chart (figure 1) the excluded patients' box should have proper grammar - should be: unknown number of examined regional nodes, unknown tumor grade or tumor size or patient race, unknown TNM or the American Joint Committee on Cancer (AJCC) stage.
Response: Thank you for this very helpful suggestion. We have revised the flow chart (Figure 1) in the manuscript according to your advice.
- Line 142 - Table 1, several variables present percentages that do not sum 100%. Some sum 100.1% others 99%. This should be corrected in order to all variables percentages sum 100%.
Response: Thank you for this comment. We have revised the Table 1 according to your advice. Thank you again for your great suggestion.
- Line 168 - Table 3, several variables present percentages that do not sum 100%. Some sum 100.1% others 99%. This should be corrected in order to all variables percentages sum 100%.
Response: Thank you for this comment. We have revised the Table 3 according to your advice. Thank you again for your great suggestion.
- Line 175 - "The multivariate Cox analysis results revealed that age ≥ 75, size >5 cm, ELNs <32, tumor located in antrum/pylorus... were associated with a poorer prognosis" - The fact that you found that patients having more distal tumors (antrum and pylorus) were associated with poorer prognosis is surprising and should be discussed, why do you believe that happened?
Response: Thank you for this comment. In this study, we found that patients having more distal tumors (antrum and pylorus) were associated with poorer prognosis believe. The reason maybe following: firstly, compared with other type of gastric cancers, gastric signet ring cell carcinoma is more aggressive, more prone to peritoneal metastasis and lymphatic infiltration, and has a poorer prognosis. Therefore, the traditional view is that total gastrectomy is usually performed for progressive gastric signet ring cell carcinoma. (BMC Cancer (2013):281 (Surgical Resection section); Cancer Manag Res (2019) :2151-2161). Moreover, compared with non-gastric signet ring cell carcinoma, gastric signet ring cell carcinoma is more likely to occur in the distal 1/3 of the stomach body (Cancer Manag Res (2019) :2151-2161),which results in a large base patients with gastric antrum/pyloric signet ring cell carcinoma. Therefore, the prognosis of distal gastric signet ring cell carcinoma is poor in this study. In the future, we can expand the sample size for further discussion. We have discussed it in the Discussion section according to your advice.
- Line 175 - "... treatment with surgery only...
Response: Thank you for this comment. We have revised this sentence in the manuscript according to your advice.
- Line 204 - The results showed that the XGBoost model that included the optimal number of ELNs (n = 32) had better predictive ability than the XGBoost model that excluded the optimal number of ELNs...
Response: Thank you for this comment. We have revised this sentence in the manuscript according to your advice.
- Line 208 - These pictures (figure 7) cannot be interpreted as the pictures are presented with a very poor definition - should be improved.
Response: Thank you for this comment. We have revised these picture (Figure 7) in the manuscript according to your advice.
- Line 225 - clearance and is an accurate method
Response: Thank you for this comment. We have revised this sentence in the manuscript according to your advice.
- Line 226 - Adequate ELNs help determine prognosis
Response: Thank you for this comment. We have revised this sentence in the manuscript according to your advice.
- Line 233 - what do you mean with "intra-operative ELNs"? Do you want to say the optimal number of dissected lymph nodes? Or the number of lymph nodes examined by the pathologist in theatre?
Response: Thank you for your comment. "Intra-operative ELNs" is to say the optimal number of dissected lymph nodes. We have revised it in the manuscript according to your advice.
- Line 236 - ...with fewer than 16 examined lymph nodes may present a poorer prognosis...
Response: Thank you for this comment. We have revised this sentence in the manuscript according to your advice.
- Line 247 - ... Currently, there are limited studies identifying the optimal number...
Response: Thank you for this comment. We have revised this sentence in the manuscript according to your advice.
- Line 250 - "...confirmed that 32 was the adequate lymph node a cut-off number for dissection, while prognosis would not be improved with ELNs >32." - This sentence is confusing, firstly should be reviewed and rewritten with a better grammar, secondly, from what I gathered, you showed exactly the opposite, that prognosis (survival) was improved with ELNs >32.
Response: Thank you very much for this careful suggestion. We have revised the manuscript according to your advice.
- Line 261 - " Finally, we used the XGBoost model compared the predict ability included the optimal number of ELNs of GSRC and excluded the optimal number of ELNs." This is a very confusing sentence and should be reviewed.
Response: Thank you for this comment. We have revised it according to your advice.
- Line 265 - What do you mean with "the best ELNs"?
Response: Thank you for this comment. "the best ELNs" means the optimal number of ELNs. We have revised it to " the optimal number of ELNs " according to your advice.
- Line 268 - However, some may believe...
Response: Thank you for this comment. We have revised the manuscript according to your advice.
- Line 274 - you mention a "large randomized clinical trial in the United States has demonstrated that systematic lymph node sampling would only prolong operative time and increase bleeding rate but has no effect on survival and does not increase postoperative hospital stay or the incidence of postoperative complications". However, your references related to this statement are 2 papers reviewing lymph node dissection for lung cancer. I believe you should not infer about abdominal lymphadenectomy presenting references of papers discussing mediastinal and hilar lymph node dissection in lung cancer.
Response: Thank you for this comment. We have revised the sentence and corrected the reference in the manuscript according to your advice.
- Line 279 - For which a further study will be conducted in the future.
Response: Thank you for this comment. We have revised the manuscript according to your advice.
- Line 282 - Asian population
Response: Thank you for this comment. We have revised the manuscript according to your advice.
- Line 282 - "gastric IMC" - Please present the meaning of abbreviations the first time you use them in the text.
Response: Thank you for this comment. We are sorry that we wrote gastric IMC by mistake. We have revised the manuscript according to your advice.

Reviewer 2 Report
Overall, this is a very interesting novel study creating a new nomogram for gastric signet ring cancer. The statistical methodology seems sound, and the results are rigorous.
My main concern/question is there is a lack of discussion with chemotherapy/radiation. It is unclear if patients received neoadjuvant chemo/radiation, adjuvant chemo/radiation, or both. I cannot find this in the methodology. The Tables list the numbers that received additional treatments to surgery, but the timing (neoadjuvant/adjuvant) is not stated (at least that I could find). This would be expected to have a big impact on outcomes (and so may need to be included in the nomogram), and this would also be expected to have a big impact on lymph node yield. If there is response to chemotherapy, then fewer lymph nodes may be identified. The authors should address this more comprehensively prior to acceptance.
There is limited study on signet ring gastric cancer, given its rarity. But authors may be able to extrapolate from other studies looking into the impact of neoadjuvant therapy on gastric cancer outcomes. Examples include PMID 27641320, 32737698, and 29127704. Otherwise, the authors should address lack of this analysis as a significant limitation in the applicability of their nomogram.
Writing should also be proofread extensively. For example, the first sentence of the abstract is not a complete sentence and is non-sensical.
Author Response
- My main concern/question is there is a lack of discussion with chemotherapy/radiation. It is unclear if patients received neoadjuvant chemo/radiation, adjuvant chemo/radiation, or both. I cannot find this in the methodology. The Tables list the numbers that received additional treatments to surgery, but the (neoadjuvant/adjuvant) is not stated (at least that I could find). This would be expected to have a big impact on outcomes (and so may need to be included in the nomogram), and this would also be expected to have a big impact on lymph node yield. If there is response to chemotherapy, then fewer lymph nodes may be identified. The authors should address this more comprehensively prior to acceptance.
Response: In this study, the data we used were obtained from the authoritative public database the Surveillance, Epidemiology, and End Results (SEER) database, using SEER*Stat software (version 8.4.0, http://seer.cancer.gov/seerstat/) from the National Cancer In-stitute. Unfortunately, in this database, only chemotherapy history is provided and information on the timing of chemotherapy and radiotherapy is missing, so this study cannot provide information on whether the treatment was neoadjuvant or adjuvant or both. Therefore, your concern about neoadjuvant and adjuvant chemotherapy and its timing cannot be shown. In this regard, we searched literature published in SCI using SEER database for similar studies, such as prognosis of bladder cancer (Front Oncol 12 (2022):789028) based on the number of lymph nodes, prognosis of lung cancer (Transl Lung Cancer Res 10(2021):815-825), prognosis of esophageal cancer (Ann Surg Oncol 27(2020):2042-2050), etc. They also did not differentiate between chemotherapy information, but did not affect the reference value of such studies. Besides, gastric signet ring cell carcinoma (GSRC) is a particular histological subtype of gastric adenocarcinomas displaying a worse prognosis. Even though the perioperative chemotherapy in resectable gastric adenocarcinomas demonstrated a significant benefit in terms of overall survival compared to surgery alone (N Engl J Med. 355 (2006):11–20; J Clin Oncol 29 (2011):1715–21), this benefit seems to be limited to non-GSRC histology (Ann Surg 254 (2011):684–93). And previous studies have suggested GSRC may have inherent chemo resistance (Eur J Cancer 30A (1994):1263–1269; Ann Surg 250 (2009):878–887; Oncol Rep 7 (2000):841–846; Clin Cancer Res 14 (2008):2012–2018), the role of chemotherapy in GSRC is controversial, especially the limited effect on positive lymph nodes. Furthermore, currently, no specific recommendation is available about the type of lymphadenectomy to perform for GSRC, and this study is the first study to investigate the optimal number of examed lymph nodes intraoperatively in GSRC. Although information on the timing of radiotherapy and chemotherapy is lacking, it is still useful for clinical practice and future studies. It needs to be acknowledged that if the study includes the information about neoadjuvant and adjuvant will further improve the application of the model constructed in this study. In this regard, we truthfully point out this flaw in the Discussion section of this study to avoid misinformation. In summary, we believe that the innovative elements of our work are more clearly emphasized, and hopefully that you could satisfactorily respond to this comment, which helped to improve the quality of the manuscript.
- There is limited study on signet ring gastric cancer, given its rarity. But authors may be able to extrapolate from other studies looking into the impact of neoadjuvant therapy on gastric cancer outcomes. Examples include PMID 27641320, 32737698, and 29127704. Otherwise, the authors should address lack of this analysis as a significant limitation in the applicability of their nomogram?
Response: Thank you for your suggestion. We have read above article carefully, unfortunately, we could only confirm if the patient has undergone chemotherapy. Although it is a type of gastric cancer, gastric signet ring cell carcinoma (GSRC) is more malignant, more aggressive, and less sensitive to chemotherapy. A retrospective study by Messager et al including 924 cases of resected GSRC was investigated to compare the survival of patients with GSRC treated with and without perioperative chemotherapy. The results showed perioperative chemotherapy provides no survival benefit in patients with GSRC (Ann Surg 254(2011):684–693). Therefore, GSRC may have a different chemosensitivity profile than non-GSRC. The prognostic impact of chemotherapy on GSRC may not be as great as that on non-GSRC. Besides, previous studies exploring the prognosis of cancer based on the number of lymph nodes also did not differentiate between chemotherapy information, but did not affect the reference value of such studies. (Front Oncol 12 (2022):789028; Transl Lung Cancer Res 10(2021):815-825). Actually, the timing of chemotherapy has influence in the survival of patients with GSRC, we truthfully point out this flaw in the Discussion section of this study to avoid misinformation according to your advice.
- Writing should also be proofread extensively. For example, the first sentence of the abstract is not a complete sentence and is non-sensical
Response: Thank you for this very helpful suggestion. We have revised the abstract section according to your advice. Besides, we have examed the manuscript carefully and sent the paper to native English-speaking colleague to improve the language. The language editorial certification is attach.

Round 2
Reviewer 1 Report
Dear authors, thank you for reviewing your manuscript.
I believe it has actually been improved.
There are still some concerns and a few points to improve.
Regarding you response to issue number 1, you claim GSRC shows a limited chemotherapy benefit ("Even though the perioperative chemotherapy in resectable gastric adenocarcinomas demonstrated a significant benefit in terms of overall survival compared to surgery alone..., this benefit seems to be limited to non-GSRC histology... And previous studies have suggested GSRC may have inherent chemo resistance..., the role of chemotherapy in GSRC is controversial, especially the limited effect on positive lymph nodes"). However you still mention in your manuscript: "...ELNs help determine prognosis and accurate staging, providing guidance for adjuvant treatment and surveillance programs following treatment. While with inadequate ELNs, N1 and N2 patients could be misdiagnosed as N0, who would receive different treatment and simpler monitoring procedures after treatment, resulting in a worse prognosis". I believe your claims are contradictory to explain why a more extended lymphadenectomy may lead to a better prognosis. I still believe your results can be interpreted in a different way and that you should mention it in your discussion.
Line 16 - "X-tile software was usedcalculated the cut- off value of ELNs..." - please correct
Line 21 - Furthermore, the predicted model based on 32 ELNs was developed and displayed as a nomogram.
Line 45 - inadequate intraoperative lymph node dissection
Line 210 - It is still difficult to interpret figure 7 due to low images resolution. Please improve it if you can.
Line 225 - "Intra-operative ELNs provide information about the thoroughness of clearance and is an accurate method to confirm the number of positive lymph nodes". - The examination of lymph nodes does not happen intra-operatively, hence the expression "Intra-operative ELNs" should be avoided. Maybe you could say: "ELNs provide information about the thoroughness of intra-operative lymph node clearance...".
Line 230 - You use reference 20 ("A large real-world cohort study of examined lymph node standards for adequate nodal staging in early non small cell lung cancer") which is related to lung cancer in order to explain a worse prognosis in gastric cancer. This should be avoided as may mislead and cause confusion, should be avoided.
Line 263 - "Finally, we used the XGBoost model compared the predict ability of models which included the optimal number of ELNs of GSRC and excluded the optimal number of ELNs." - This sentence is confusing and should be improved.
Line 271 - "...we found that patients having more distal tumors (antrum and pylorus) were associated with poorer prognosis. The reason maybe following: firstly, compared with other type of gastric cancers, gastric ring cell carcinoma is more aggressive, more prone to peritoneal metastasis and lymphatic infiltration, and has a poorer prognosis. Therefore, the traditional view is that total gastrectomy is usually performed for progressive GSRC. Moreover, compared with non-GSRC, GSRC is more likely to occur in the distal 1/3 of the stomach body, which results in a large base patients with gastric antrum/pyloric signet ring cell carcinoma. Therefore, the prognosis of distal GSRC is poor in this study". Your cohort of patients only presented GSRC, hence the fact that that a specific histological type of tumor is more frequent in one location does not explain why it shows worse prognosis at that same location, when comparing with the same histology type at different locations. This should be better explained and discussed.
Line 302 - Therefore, there may be diversity in the way lymph nodes are examined and processed.
Author Response
Responses to the reviewer’s comments
Reviewer #1:
- Regarding you response to issue number 1, you claim GSRC shows a limited chemotherapy benefit ("Even though the perioperative chemotherapy in resectable gastric adenocarcinomas demonstrated a significant benefit in terms of overall survival compared to surgery alone..., this benefit seems to be limited to non-GSRC histology... And previous studies have suggested GSRC may have inherent chemo resistance..., the role of chemotherapy in GSRC is controversial, especially the limited effect on positive lymph nodes"). However you still mention in your manuscript: "...ELNs help determine prognosis and accurate staging, providing guidance for adjuvant treatment and surveillance programs following treatment. While with inadequate ELNs, N1 and N2 patients could be misdiagnosed as N0, who would receive different treatment and simpler monitoring procedures after treatment, resulting in a worse prognosis". I believe your claims are contradictory to explain why a more extended lymphadenectomy may lead to a better prognosis. I still believe your results can be interpreted in a different way and that you should mention it in your discussion.
Response: Thank you for your kindly comment. We apologize that we may not have made it clear on the issue number 1 that the impact of preoperative chemotherapy on reducing the number of positive lymph nodes in patients with GSRC may be not significant. Hence, the distinction between neoadjuvant or adjuvant chemotherapy on determining the optimal number of intraoperative lymph nodes in patients with GSRC, although it may have an impact, might not affect the reference value of our study. Determining the optimal number of examed lymph nodes can help physicians to stage patients with GSRC more accurately and thus guide subsequent more accurate treatment. We have revised the statement in the manuscript according to your advice. Thank you again for your helpful comment, which can improve the quality of our manuscript.
- Line 16 - "X-tile software was usedcalculated the cut- off value of ELNs..." - please correct
Response: Thank you for this helpful comment. We have corrected the sentence according to your advice.
- Line 21 - Furthermore, the predicted model based on 32 ELNs was developed and displayed as a nomogram.
Response: Thank you very much for this careful suggestion. We have revised it in the manuscript according to your advice. We really appreciate your patient and careful advice
- Line 45 - inadequate intraoperative lymph node dissection
Response: Thank you for your kindly comment. We have revised it in the manuscript according to your advice.
- Line 210 - It is still difficult to interpret figure 7 due to low images resolution. Please improve it if you can.
Response: Thank you for your helpful comment. We have revised the figure in the manuscript according to your advice. If you still feel it is difficult to interpret figure 7, please fell free to contact us. Thank you again for your suggestion.
- Line 225 - "Intra-operative ELNs provide information about the thoroughness of clearance and is an accurate method to confirm the number of positive lymph nodes". - The examination of lymph nodes does not happen intra-operatively, hence the expression "Intra-operative ELNs" should be avoided. Maybe you could say: "ELNs provide information about the thoroughness of intra-operative lymph node clearance...".
Response: Thank you for your kindly and patient comment. We have revised it in the manuscript according to your advice.
- Line 230 - You use reference 20 ("A large real-world cohort study of examined lymph node standards for adequate nodal staging in early non small cell lung cancer") which is related to lung cancer in order to explain a worse prognosis in gastric cancer. This should be avoided as may mislead and cause confusion, should be avoided.
Response: Thank you for your kindly comment. We have corrected the reference in the manuscript according to your advice.
- Line 263 - "Finally, we used the XGBoost model compared the predict ability of models which included the optimal number of ELNs of GSRC and excluded the optimal number of ELNs." - This sentence is confusing and should be improved.
Response: Thank you for your kindly comment. We have revised it in the manuscript according to your advice.
- Line 271 - "...we found that patients having more distal tumors (antrum and pylorus) were associated with poorer prognosis. The reason maybe following: firstly, compared with other type of gastric cancers, gastric ring cell carcinoma is more aggressive, more prone to peritoneal metastasis and lymphatic infiltration, and has a poorer prognosis. Therefore, the traditional view is that total gastrectomy is usually performed for progressive GSRC. Moreover, compared with non-GSRC, GSRC is more likely to occur in the distal 1/3 of the stomach body, which results in a large base patients with gastric antrum/pyloric signet ring cell carcinoma. Therefore, the prognosis of distal GSRC is poor in this study". Your cohort of patients only presented GSRC, hence the fact that that a specific histological type of tumor is more frequent in one location does not explain why it shows worse prognosis at that same location, when comparing with the same histology type at different locations. This should be better explained and discussed.
Response: Thank you very much for your careful and patient comment. After careful examination of our data, we found that we confused the dummy variables set during the cox regression analysis, so we ended up with the error that gastric antrum/pylorus was a risk factor, for which we are sorry. We have now corrected the corresponding content in the manuscript and checked our results again carefully according to your advice. In summary, we believe that the quality of our work are more clearly emphasized, and hopefully that you could satisfactorily respond to this comment, which helped to improve the quality of the manuscript.
- Line 302 - Therefore, there may be diversity in the way lymph nodes are examined and processed.
Response: Thank you for your kindly and patient comment. We have revised it in the manuscript according to your advice.
Reviewer 2 Report
Authors have addressed concerns with new sections, particularly expounding upon study limitations.
Author Response
Thank you for your kindly comment. In summary, we believe that the innovative elements of our work are more clearly emphasized, and hopefully with your comment.